# Knowledge and attitude of populations on blackflies and onchocerciasis and participation in mass drug administration in first-line communities near Erin-Ijesha and Arinta waterfalls, Southwest Nigeria

Oluwadamilare Ganiu Dauda[1]*, Akinlabi Mohammed Rufai[1], Olabanji Ahmed Surakat[1], Oluwatoyin Adeola Oyeniran[2], Olusegun Quadri Adeshina[1], Abuh Ojomona Oboro[1], Azeezat Ayanbukola Ayannibi[3], Rhidollah Folashade Oyewusi[1], Glory Blessing Jokanola[1], Kamilu Ayo Fasasi[1], Monsuru Adebayo Adeleke[1]

**1** Department of Animal and Environmental Biology, Osun State University, Osogbo, Osun State, Nigeria, **2** Department of Medical Microbiology and Parasitology, Osun State University Teaching Hospital, Osogbo, Osun State, Nigeria, **3** Department of Pure and Applied Biology, Ladoke Akintola University of Technology, Ogbomoso, Oyo State, Nigeria

* dare_dauda@yahoo.com

## Abstract

### Background

Preliminary assessments have identified blackfly biting activity at Erin-Ijesha and Arinta waterfalls in Southwest Nigeria, predisposing first-line communities to a high risk of onchocerciasis. There is a need to assess knowledge of blackflies, onchocerciasis, and participation in treatment programs among residents of first-line communities near the waterfalls to evaluate onchocerciasis transmission risk.

### Methods

A cross-sectional survey was conducted among 236 residents at Erin-Ijesha and Ipole-Iloro, two first-line communities using structured questionnaires. Data were entered into Microsoft Excel and analyzed using the Statistical Package for the Social Sciences (SPSS) software version 21. Relationships between variables were analyzed using t-test and chi-square, with a 95% confidence interval.

### Results

Majority (73.3%) in Erin-Ijesha and (83.7%) in Ipole-Iloro reported knowledge of blackflies, with 83.3% and 87.0% aware of onchocerciasis, respectively. The majority of participants had no knowledge of how onchocerciasis is transmitted, with only 8.7% and 10.5% in Erin-Ijesha and Ipole-Iloro, respectively, linking the disease to blackfly bites. Ivermectin uptake was relatively high at 75.7% and 76.7% in

**Data availability statement:** All data generated and/or analysed during this study is uploaded as Supporting information.

**Funding:** The author(s) received no specific funding for this work.

**Competing interests:** The authors have declared that no competing interests exist.

Erin-Ijesha and Ipole-Iloro, respectively. Willingness to participate in future treatment programs was lower, at 64.7% and 64.0% in Erin-Ijesha and Ipole-Iloro, respectively. Level of education significantly influenced willingness to participate in future mass distribution of ivermectin ($p < 0.05$).

## Conclusion

The limited understanding of blackfly bioecology and onchocerciasis among residents may lead to increased exposure to bites, thereby raising the risk of transmission. To address this, the federal and state ministries of health, along with treatment implementing partners, should enhance sensitization efforts and public health education during the annual Mass Drug Administration campaigns. Improving knowledge in these high-risk communities will encourage residents to take preventive measures against bites and improve treatment uptake, ultimately reducing the risk of disease transmission.

### Author summary

Preliminary assessments have identified blackfly biting activity at Erin-Ijesha and Arinta waterfalls in Southwest Nigeria, predisposing first-line communities to a high risk of onchocerciasis. These blackflies make use of fast flowing rivers for the development of their immature stages and the adult fly transmits the parasite *Onchocerca volvulus* that causes severe skin and eye disorder. The disease is particularly prevalent in communities near suitable river systems. Erin-Ijesha and Ipole-Iloro, two first-line communities near the waterfalls, have been involved in mass ivermectin distribution programs for years. However, this study reveals a striking lack of knowledge among residents regarding the bioecology of blackflies, severity of onchocerciasis and the importance of ivermectin uptake. This knowledge gap has contributed to suboptimal treatment coverage, with some individuals refusing to participate in annual mass drug distribution. Given the high-risk status of these communities, these findings highlight the urgent need for the federal and state ministries of health and the NTD implementing partners to intensify on community sensitization efforts before and during mass drug administration campaigns to address misconceptions, improve public health awareness and optimize treatment program outcomes.

## Introduction

Onchocerciasis is a skin disfiguring and blinding tropical infectious disease that significantly impacts affected communities and caused by the filarial nematode *Onchocerca volvulus* and transmitted by *Simulium* vector [1]. The epidemiology of onchocerciasis is influenced by the abundance, biting behaviour and vectorial capacity of local blackfly populations. The disease is found in isolated regions of Latin

America and Yemen but the overwhelming majority of the global burden of onchocerciasis is concentrated in Africa, with more than 99% of cases occurring in 31 countries within sub-Saharan Africa [1]. Nigeria holds the highest global burden of onchocerciasis, accounting for approximately 40% of all cases worldwide [2,3]. The disease manifests through skin lesions accompanied by intense itching, as well as severe eye damage that can lead to blindness [1,4]. It is actually the second leading cause of preventable blindness in the world behind trachoma [5].

In Africa, onchocerciasis intervention programs shifted their goal from merely managing the disease as a public health issue by achieving a reduction in skin and ocular complications to targeting the elimination of the disease [6]. This change was driven by evidence from several studies demonstrating that *O. volvulus* elimination is achievable through community-directed ivermectin treatment (CDTI) [7–9]. Based on this evidence, disease control efforts in Nigeria have also primarily focused on mass drug administration (MDA) with ivermectin, a strategy that has been implemented for nearly 20 years, with states progressing at varying stages toward elimination [2]. This approach is crucial as it empowers endemic communities to decide the timing of drug distribution, choose their distributors, determine the distribution method, and take part in supervising mass drug administration [6]. Currently, ivermectin has been administered to over 34 million Nigerians achieving 80.6% geographical coverage [10].

A preliminary survey of rivers in Southwest Nigeria revealed blackfly activity at two major tourist sites, the Erin-Ijesha and Arinta waterfalls in Osun and Ekiti States, respectively, where such biting activity had not been previously reported or documented. Both states were classified as mesoendemic for onchocerciasis based on several surveys using the Rapid Epidemiological Mapping of Onchocerciasis (REMO) mapping conducted in 1995–1996 [2] and have been receiving ivermectin treatment since 2000 [11]. Both states are considered to be on track for disease elimination [2]. However, there is no evidence-based information on prevalence status or the infectivity status of blackflies at these sites in order to determine the transmission risk of onchocerciasis in first-line communities nearest to the waterfalls. The absence of data on onchocerciasis prevalence in these two regions, despite the observed blackfly activity, raises concerns about potential transmission dynamics, given the vector's role in the epidemiology of the disease. This study was designed as an epidemiological survey to evaluate community knowledge, attitudes, and determine level of participation in ivermectin mass drug administration (MDA) in order to assess the onchocerciasis transmission risk in the region. As Nigeria transitions from control to elimination of onchocerciasis, this data will be vital for the Federal and State Ministries of Health, the National Onchocerciasis Elimination Committee (NOEC), and treatment implementing partners in developing targeted strategies to ensure optimal therapeutic coverage and achieve disease elimination.

## Materials and methods

### Ethics statement

The ethical approval for this study was granted by the Ethics Review Committees of the Osun State Ministry of Health (reference number OSHREC/PRS/569T/334) and the Ekiti State Ministry of Health (reference number MOH/EKHREC/EA/P/50). Written informed consent was first obtained from community leaders and religious heads of the communities. For participants aged 10 and below 18 years, recruitment was conducted with the written consent of their parents or guardians after explaining the study's objectives in English and in the local language (Yoruba). Written informed consent was also obtained from participants above 18 years. All participants were informed that participation was voluntary and that they could withdraw at any point without any negative consequences to them. The research adhered to international ethical standards, including the 1964 Helsinki Declaration and its subsequent amendments.

### Study area

This study was conducted between October and November 2024 in two first-line communities near the Arinta and Erin-Ijesha waterfalls. The first site, Erin-Ijesha (N7.569368, E4.891340), located in Oriade Local Government Area, Osun

State, is the nearest community to the Erin-Ijesha waterfall. The second site, Ipole-Iloro (N7.586372, E4.936383), situated in Ekiti West Local Government Area, Ekiti State, is the first-line community to the Arinta waterfall. There is no documented information on the level of onchocerciasis endemicity in the LGAs where these communities are located. However, blackfly activity observed during a preliminary assessment of the waterfalls close to the communities prompted the selection of these locations for the study (Fig 1).

This map was created using primary data collected by the authors and developed in ArcGIS v10.8 software. Shapefiles for Nigeria, including boundary polygons and water bodies across all administrative levels, were sourced from publicly accessible databases available at Humanitarian Data Exchange, and are licensed under CC BY 4.0 ([https://data.humdata.org/](https://data.humdata.org/)). No changes were made to the original shapefiles. The authors grant permission for the reuse of this map without restriction, provided appropriate credit is given to the source.

## Study design and determination of sample size

This community-based cross-sectional study was conducted in two purposively selected first-line communities between October 2024 and November 2024. Local neglected tropical disease coordinators (LNTDs), facility health workers from the community primary health centers (PHCs), and local mobilizers facilitated community sensitization and mobilization efforts. Participants aged 10 years and above and who have been resident in the communities for 5 years and above were enrolled in the study. In the absence of prior data on prevalence and community knowledge of blackflies and onchocerciasis, Cochran's formula; $n = (Z^2 * P(1-P))/ E^2$ was used to determine sample size. Where n = the required

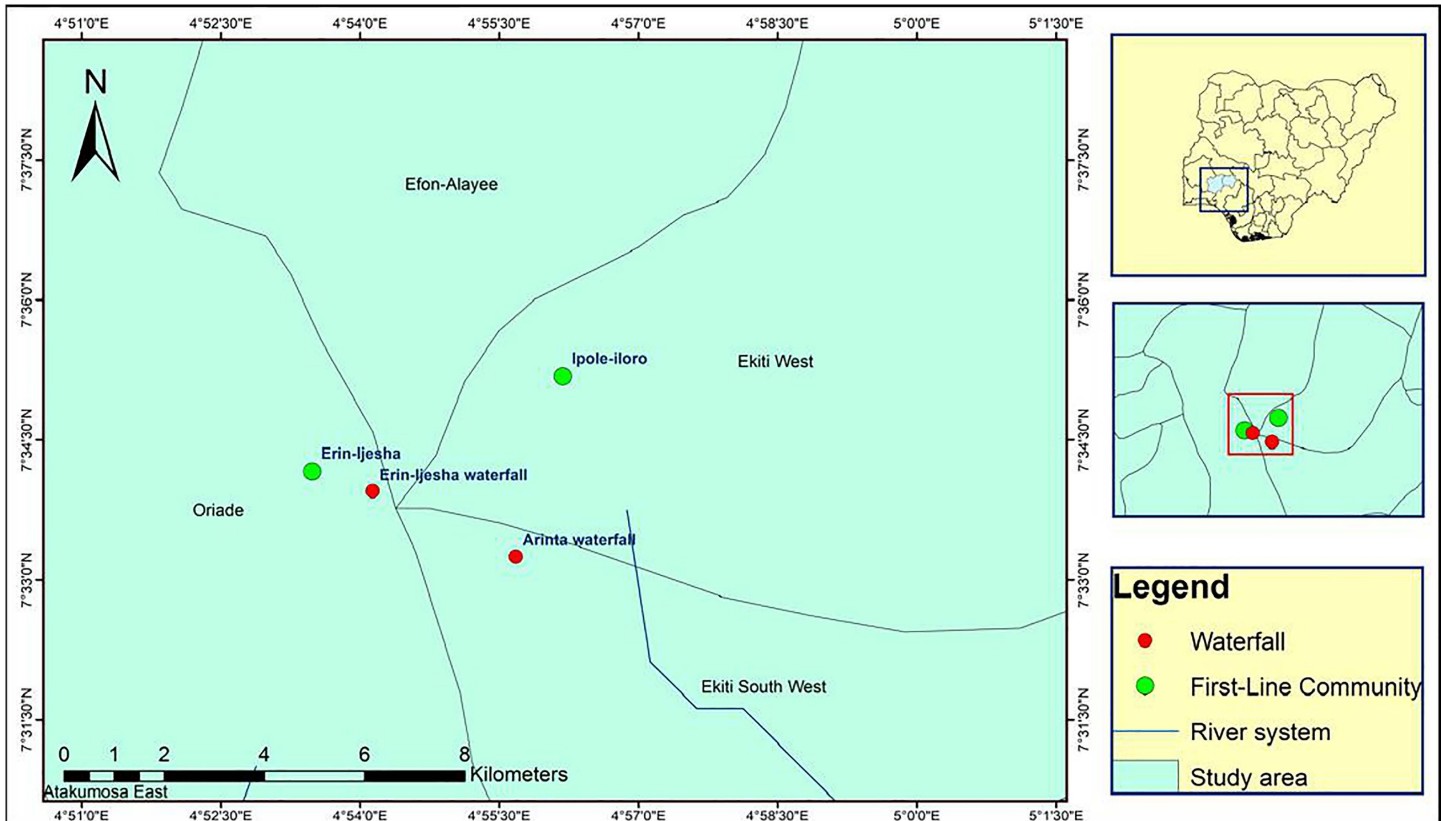

**Fig 1. Map showing study locations and the waterfalls.**

sample size, Z = the standard normal deviate, usually set at 1.96 for a 95% confidence level, P = the expected population with knowledge of onchocerciasis in the population, which is usually assumed to be 0.5 (50%) in the absence of any previous record and E = the desired level of precision, which is usually set at 0.05 (5%) for most studies. The estimated sample size was 385.

## Data collection and analysis

Following informed consent, four trained personnel administered the questionnaire with assistance from a local guide who translated the questions into the local language to ensure clarity and comprehension. The questionnaire comprised four sections: (i) socio-demographic information, (ii) knowledge of blackflies, (iii) knowledge of onchocerciasis, and (iv) attitudes and participation in mass ivermectin administration. Data were entered into Microsoft Excel and analyzed using the 2021 version of the Statistical Package for the Social Sciences (SPSS). Descriptive statistics were performed, and relationships between variables were assessed using t-tests and chi-square tests, with proportions analyzed at a 95% confidence interval.

## Results

### Socio-demographic characteristics of study participants

A total of 236 respondents from the two communities participated in the study, with 150 individuals from Erin-Ijesha and 86 from Ipole-Iloro. Female participants accounted for the majority (78.4%), while males constituted 21.6%. The largest age group represented was 61–80 years (44.1%) while the lowest age group represented were between the ages 10–20 (4.6%). Respondents with no formal education comprised the highest proportion (34.3%), whereas those with tertiary education were the least represented (7.7%) (Table 1).

### Knowledge on blackfly and bioecology of blackfly

Knowledge about blackflies varied among participants in the two communities. In Erin-Ijesha, 110 (73.3%) of respondents reported having heard of blackflies, while 40 (26.7%) had no knowledge. Additionally, 105 (70.0%) identified blackflies as a biting nuisance in the community, whereas 45 (30.0%) were unaware of this activity. In Ipole-Iloro, 72 (83.7%) of participants were familiar with blackflies, compared to 14 (16.3%) who have no knowledge. Furthermore, 70 (81.4%) confirmed the biting activity of blackflies in the community, while 16 (18.6%) lacked awareness. When prompted about the origin of blackflies, 81 (54.0%) of respondents in Erin-Ijesha attributed their presence to trees, 56 (37.3%) had no idea, and only 3 (2.0%) correctly identified rivers as their source. Similarly, in Ipole-Iloro, 63 (73.3%) believed blackflies originated from trees, 21 (24.4%) were unsure, and only 2 (2.3%) recognized rivers as their breeding sites. Occupation and their level of education were not found to have any significant influence on knowledge on bioecology of blackflies in the two communities ($p > 0.05$) (Table 2).

### Knowledge on onchocerciasis

At Erin-Ijesha, 125 (83.3%) of respondents demonstrated knowledge and awareness of onchocerciasis, with 88.0% of the respondents accurately identifying the local name for the disease in their community. Similarly, at Ipole-Iloro, 74 (86.0%) of participants were familiar with the disease, and 75 (87.2%) of community members correctly recognized its local name. Despite high awareness levels of the disease, knowledge about the mode of transmission of onchocerciasis was notably poor in both communities. At Erin-Ijesha, 76.7% of respondents have no idea of the disease's transmission route, while at Ipole-Iloro, 82.5% of the respondents also had no idea. Only 8.7% and 10.5% of respondents in Erin-Ijesha and Ipole-Iloro, respectively, correctly identified the blackfly as the vector responsible for the transmission. Misconceptions about the disease were prevalent, and knowledge of its common symptoms was limited. In Erin-Ijesha, 40.0% of respondents

**Table 1. Socio-demographic characteristics of participants.**

| Variables | No. of individuals at Erin-Ijesha (%) | No. of individuals at Ipole-Iloro (%) | Total number (%) |
|---|---|---|---|
| Number of participants | 150 | 86 | 236 |
| **Age** | | | |
| 10–20 | 8 (5.3) | 3 (3.5) | 11 (4.6) |
| 21–40 | 32 (21.3) | 4 (4.7) | 36 (15.3) |
| 41–60 | 58 (38.7) | 27 (31.4) | 85 (36.0) |
| 61–80 | 52 (34.7) | 52 (60.4) | 104 (44.1) |
| **Sex** | | | |
| Male | 31 (20.7) | 20 (23.3) | 51 (21.6) |
| Female | 119 (79.3) | 66 (76.7) | 185 (78.4) |
| **Education** | | | |
| Primary | 34 (24.7) | 35 (40.7) | 69 (29.2) |
| Secondary | 55 (36.7) | 13 (15.1) | 68 (28.8) |
| Tertiary | 12 (8.0) | 6 (7.0) | 18 (7.7) |
| No formal education | 49 (32.6) | 32 (37.2) | 81 (34.3) |
| **Occupation** | | | |
| Student | 10 (6.7) | 3 (3.5) | 13 (5.5) |
| Farmer | 33 (22.0) | 41 (47.7) | 74 (31.4) |
| Trader | 77 (51.3) | 12 (14.0) | 89 (37.7) |
| Civil Servant | 6 (4.0) | 6 (6.9) | 12 (5.1) |
| Retired | 15 (10.0) | 10 (11.6) | 25 (10.6) |
| Unemployed | 9 (6.0) | 14 (16.3) | 23 (9.7) |

reported having no knowledge of the symptoms, while in Ipole-Iloro, this lack of knowledge was observed in 41.9% of the respondents (Table 3).

## Attitude and level of participation in mass drug administration (MDA)

Awareness of ivermectin was high in both communities, with 127 (84.7%) in Erin-Ijesha and 73 (84.9%) in Ipole-Iloro having heard about the drug. Among those aware, 113 (75.3%) in Erin-Ijesha and 66 (77.6%) in Ipole-Iloro reported receiving at least one round of ivermectin. When asked about the total number of ivermectin rounds received, 23 (15.3%) of respondents in Erin-Ijesha reported receiving one round, 31 (20.7%) two rounds, 30 (20.0%) more than two rounds, and 29 (19.3%) more than five rounds. At Ipole-Iloro, 16 (24.2%) had received one round, 11 (16.7%) two rounds, 22 (33.3%) more than two rounds, and 17 (25.8%) more than five rounds. Among those 37 (24.7%) and 20 (23.3%) respondents in Erin-Ijesha and Ipole-Iloro respectively who had not received ivermectin, 11 (29.7%) in Erin-Ijesha and 8 (40.0%) in Ipole-Iloro cited refusal as the reason, while 10 (27.0%) and 9 (45.0%), respectively, reported unawareness of the drug distribution. Regarding willingness to participate in future ivermectin campaigns, 97 (64.7%) of respondents in Erin-Ijesha and 55 (64.0%) in Ipole-Iloro expressed willingness, whereas 53 (35.3%) and 31 (36.0%) respectively, declined.

In Erin-Ijesha, 74 (49.3%) respondents reported being aware of community sensitization efforts on onchocerciasis, while 76 (50.7%) indicated a lack of awareness. Similarly, at Ipole-Iloro, 47 (54.7%) were aware of such efforts, while 39 (45.3%) were not. When asked if sensitization efforts on onchocerciasis elimination could be improved, 116 (77.3%) in Erin-Ijesha and 77 (89.5%) in Ipole-Iloro responded affirmatively. Regarding suggestions for improvement in sensitization efforts in Erin-Ijesha, 44 (29.3%) recommended increased health worker outreach, while 12 (8.0%) suggested leveraging radio, TV, and social media campaigns. In Ipole-Iloro, 13 (15.1%) similarly advocated for more health worker outreach,

**Table 2. Knowledge on blackflies and bioecology.**

| Questions | Response | Erin-Ijesha (%) | Ipole-Iloro (%) |
|---|---|---|---|
| Have you heard of blackflies | Yes | 110 (73.3) | 72 (83.7) |
| | No | 40 (26.7) | 14 (16.3) |
| Do blackflies bite in your community? | Yes | 105 (70.0) | 70 (81.4) |
| | No | 45 (30.0) | 30 (34.9) |
| Where do these blackflies primarily come from? | Trees | 81 (54.0) | 63 (73.3) |
| | Rivers | 3 (2.0) | 2 (2.3) |
| | Others | 10 (6.7) | 0 (0) |
| | No idea | 56 (37.3) | 21 (24.4) |
| Which part of the body do blackflies mostly bite? | Head | 1 (0.7) | 1 (1.2) |
| | Hand | 16 (10.7) | 6 (6.9) |
| | Legs | 86 (57.3) | 62 (72.1) |
| | No idea | 47 (31.3) | 17 (19.8) |
| What are the immediate effects of a blackfly bite? | Itching | 97 (64.7) | 66 (76.7) |
| | Swelling | 5 (3.3) | 0 (0.0) |
| | No idea | 48 (32.0) | 20 (23.3) |
| During which season do blackflies bite more frequently? | Dry | 69 (46.0) | 46 (53.5) |
| | Rainy | 81 (54.0) | 40 (46.5) |
| What time of day do blackflies usually bite? | Day | 75 (50.0) | 63 (73.3) |
| | Night | 27 (18.0) | 5 (5.8) |
| | No idea | 48 (32.0) | 18 (20.9) |
| How do you prevent blackflies from biting you? | Wearing socks | 11 (7.3) | 7 (8.1) |
| | Clothings | 68 (45.3) | 37 (43.0) |
| | Leaf extract | 1 (0.7) | 1 (1.2) |
| | Ointments | 8 (5.3) | 6 (7.0) |
| | Nothing | 62 (41.3) | 35 (40.7) |

and 13 (15.1%) also supported the use of radio, TV, and social media for sensitization campaigns. In Erin-Ijesha, education level was found to significantly influenced willingness to participate in future mass administration of ivermectin (p = 0.00). Additionally, significant association was also observed between educational level and willingness to participate in future mass administration of ivermectin in Ipole-Iloro (p = 0.04). (Table 4).

## Discussion

This study was carried out in the study communities based on outcome of a preliminary breeding site assessment of rivers in Southwest Nigeria that revealed the biting activity of blackflies at the Erin-Ijesha and Arinta waterfall. The high biting density observed at the waterfalls necessitated the need to conduct a knowledge, attitude and practice of community members with regards to blackfly and its ecology, onchocerciasis and level of participation in treatment programs in first-line communities close to the waterfalls to determine the transmission risk of onchocerciasis in the communities.

Majority of participants in both communities demonstrated awareness of blackflies with most acknowledging the biting nuisance they cause. However, despite this impressive knowledge, the majority lacked an understanding of the bioecology of blackflies. Misconceptions were widespread, with only 2.0% of respondents in Erin-Ijesha and 2.3% in Ipole-Iloro correctly identifying that blackflies breed in fast-flowing rivers, while the majority believed that the flies originate from trees. This misunderstanding aligns with previously documented misconceptions [12–14]. The lack of knowledge about blackfly bioecology among residents in endemic communities may stem from onchocerciasis control efforts historically focusing on

**Table 3. Knowledge of participants on onchocerciasis.**

| Questions | Response | Erin-Ijesha (%) | Ipole-Iloro (%) |
|---|---|---|---|
| Do you know what onchocerciasis is? | Yes | 125 (83.3) | 74 (86.0) |
| | No | 25 (16.7) | 12 (14.0) |
| What is the local name for onchocerciasis in your community? | Sobia (Guinea worm) | 0 (0.0) | 0 (0.0) |
| | Naarun (Onchocerciasis) | 133 (88.7) | 75 (87.2) |
| | Arun fopawon (Soil-transmitted helminths) | 0 (0.0) | 0 (0.0) |
| | No idea | 17 (11.3) | 11 (12.8) |
| How is the disease transmitted? | Mosquito bite | 1 (0.7) | 3 (3.5) |
| | Blackfly bite | 13 (8.7) | 9 (10.5) |
| | Witchcraft | 0 (0.0) | 0 (0.0) |
| | Hereditary | 21 (14.0) | 3 (3.5) |
| | No idea | 115 (76.7) | 71 (82.5) |
| What are the common symptoms of onchocerciasis? | Nodules | 12 (8.0) | 9 (10.5) |
| | Leopard skin | 70 (46.7) | 18 (20.9) |
| | Blindness/Loss of vision | 8 (5.3) | 23 (26.7) |
| | No idea | 60 (40.0) | 36 (41.9) |
| Are you aware that this disease can lead to blindness if not treated? | Yes | 69 (46.0) | 40 (46.5) |
| | No | 81 (54.0) | 46 (53.5) |

parasite through mass drug administration, with little to non-existent vector control implementation in these communities. This poor knowledge could inadvertently increase exposure to blackfly bites, thereby further increasing transmission risk of onchocerciasis in these communities.

There were variations in participants' knowledge of seasonal blackfly biting activity across the study communities. In Erin-Ijesha, 54% of participants reported that blackflies bite more during the wet season while in Ipole-Iloro, 53.5% believed biting intensity is higher in the dry season. This contrasts with other studies, such as [13] where most participants associated higher biting frequency with the wet season. These variations may stem from regional differences in vector dynamics. Some studies have reported greater blackfly densities during dry season than wet season [15,16], attributed to ecological and climatic factors specific to different *Simulium* species in different ecological zones. Majority of the respondents identified itching as an immediate effect of blackfly bites. This after-effect occurs when the flies inject saliva containing irritant compounds from the epidermal to the dermal layer of the skin, triggering histamine receptors and causing itching along with other skin responses [14,17]. In response to the painful bites of blackflies, most respondents reported using protective measures such as wearing clothing, socks, and applying ointments. These practices align with findings from previous studies conducted across Africa [4,13,18]. Adeleke et al. [12] also documented the use of chemical mixtures, including diesel oil, palm oil, kerosene, and leaf extracts, as repellents. While effective, some of these substances pose health risks when applied to the skin or inhaled during application, potentially causing severe headaches, respiratory difficulties, skin irritation, or burns [14]. Although many take precautionary measures against the flies, a significant proportion of respondents, 41.3% in Erin-Ijesha and 40.7% in Ipole-Iloro reported taking no measures to prevent bites. This lack of protective behaviour leaves a substantial number of the population exposed to bites, thereby increasing the risk of onchocerciasis transmission within these communities.

It is noteworthy that the majority of participants demonstrated awareness of onchocerciasis and correctly identified its local name, "naarun." Similar patterns of knowledge have been observed in other studies, where most respondents recognized the disease and provided its local name [19,20]. This basic knowledge is likely attributable to the mass drug administration (MDA) campaigns with ivermectin, which have been ongoing in these communities for over two decades.

**Table 4. Attitude and level of participation in mass drug administration (MDA).**

| Questions | Response | Erin-Ijesha (%) | Ipole-Iloro (%) |
|---|---|---|---|
| Have you heard about ivermectin or Mectizan before? | Yes | 127 (84.7) | 73 (84.9) |
| | No | 23 (15.3) | 13 (15.1) |
| Have you ever received ivermectin before? | Yes | 113 (75.3) | 66 (76.7) |
| | No | 37 (24.7) | 20 (23.3) |
| If Yes, how many rounds? | once | 23 (15.3) | 16 (24.2) |
| | Twice | 31 (20.7) | 11 (16.7) |
| | > 2 times | 30 (20.0) | 22 (33.3) |
| | > 5 times | 29 (19.3) | 17 (25.8) |
| If you have not received treatment, what is the reason? | Not aware of distribution | 10 (27.0) | 9 (45.0) |
| | I just refuse | 11 (29.7) | 8 (40.0) |
| | Not always around | 4 (10.8) | 2 (10.0) |
| | Others | 12 (32.4) | 1 (5.0) |
| Do you know of any local medication or treatment for onchocerciasis? | Yes | 36 (24.0) | 14 (16.3) |
| | No | 114 (76.0) | 72 (83.7) |
| Are you aware of any community sensitization efforts regarding onchocerciasis? | Yes | 74 (49.3) | 47 (54.7) |
| | No | 76 (50.7) | 39 (45.3) |
| Do you believe efforts to sensitize the community about onchocerciasis elimination can be improved? | Yes | 116 (77.3) | 77 (89.5) |
| | No | 4 (2.7) | 1 (1.2) |
| | Unsure | 30 (20.0) | 8 (9.3) |
| What would help increase awareness and uptake of ivermectin? | More health worker outreach | 44 (29.3) | 13 (15.1) |
| | Radio/TV/social media campaigns | 12 (8.0) | 13 (15.1) |
| | Educational sessions in schools | 10 (6.7) | 4 (4.7) |
| | Others | 24 (16.0) | 7 (8.1) |
| | No response | 60 (40.0) | 49 (56.9) |
| Have you noticed any stigma or discrimination toward those who show symptoms of onchocerciasis in your community? | Yes | 6 (4.0) | 0 (0.0) |
| | No | 144 (96.0) | 86 (100) |
| Would you be willing to participate in future MDA for onchocerciasis? | Yes | 97 (64.7) | 55 (64.0) |
| | No | 53 (35.3) | 31 (36.0) |

However, despite this encouraging disease awareness, most participants, 76.7% in Erin-Ijesha and 82.5% in Ipole-Iloro were unaware of how onchocerciasis is transmitted. Only 8.7% and 10.5% in Erin-Ijesha and Ipole-Iloro, respectively, associated the disease with blackfly bites. This limited knowledge of the vector is a reflection of the insufficient community sensitization and education efforts by the community drug distributors (CDDs). This low knowledge can contribute to increased transmission risk as individuals are less likely to take preventive measures when unaware of the risks posed by blackfly exposure. Additionally, many participants reported no knowledge of the disease's symptoms or its potential to cause blindness. Comparable findings have been documented in other southern regions of Nigeria, where study participants failed to link the ocular symptoms of onchocerciasis to blackfly bites [19–21]. The lack of knowledge of these late-stage manifestations could potentially impact drug acceptability and uptake, thereby further increasing transmission risk. These findings further indicate shortfall in community health education and sensitization by CDDs during mass drug administration campaigns.

A notable finding from this study is the high percentage of individuals in both communities who were aware of and had received ivermectin. Majority of participants were aware of ivermectin with 75.3% in Erin-Ijesha and 76.7% in Ipole-Iloro reporting they had received at least one round of treatment. A possible explanation for the high uptake of ivermectin despite low knowledge of onchocerciasis could be due to the familiarity of community members with the CDDs, who are often known members of the community. This familiarity likely encourages ivermectin uptake, further emphasizing the effectiveness of engaging community members as CDDs. Additionally, many respondents claimed ivermectin also treats intestinal infections. This aligns with findings previously reported in Ethiopia [4]. Among those who had not participated in the treatment programs, many were persistent drug refusers, citing no specific reasons for rejecting the medication. Majority of these people who simply refuse the medicine in both communities had no formal education, further highlighting the importance of education on treatment uptake. Others were either unaware of the distribution efforts or unavailable during the campaigns. These findings suggest that community drug distributors rarely follow up with individuals who miss treatment, limiting both treatment uptake and therapeutic coverage. Similar observations were reported by [20] where individuals excluded were not often revisited, hindering efforts to achieve inclusive coverage. These untreated individuals, who remain as exposed to blackfly bites as those on ivermectin, may act as reservoir hosts and a source of re-infection of onchocerciasis within these communities [22]. The limited understanding of the disease's symptoms contributes to inappropriate alternative treatment-seeking behaviours [23]. Some of the participants claimed to take local herbs for the treatment of onchocerciasis. This is similar to what was reported in Southern Nigeria [20]. Conversely, improved awareness is associated with better prevention measures, adherence to treatment, and reduced economic burden [20]. There was no significant relationship between knowledge of onchocerciasis and ivermectin uptake in either community (p > 0.05).

The lack of adequate sensitization again became evident when participants were asked about their willingness to engage in future mass drug administration programs. Only 64% in Erin-Ijesha and 64.7% in Ipole-Iloro expressed a willingness to participate. However, education was found to significantly influence this decision in both communities, with higher levels of education correlating with greater acceptance of ivermectin treatment (p < 0.05). This further highlights the importance of education in improving treatment acceptance, suggesting that more health education in schools, targeted health workers outreach and media sensitization could significantly improve acceptability and uptake of ivermectin in endemic communities.

## Conclusion

The study reveals limited knowledge of blackfly bioecology, onchocerciasis, and its public health implications among residents of both first-line communities. This knowledge gap poses a significant challenge to ongoing control efforts and may hinder progress toward disease elimination. Given the high-risk status of these communities due to proximity to active breeding sites, it is important for federal and state health ministries and their treatment implementing partners, to prioritize targeted interventions. Improving sensitization campaigns and enhancing community awareness are crucial steps toward increasing treatment uptake, with a view to reducing the transmission risk of onchocerciasis and accelerating progress toward disease elimination

## Supporting information

**S1 File. Structured Questionnaire used for the survey of knowledge, practices and level of participation in mass drug administration at Erin-Ijesha and Ipole-Iloro communities in Southwest, Nigeria.**
(DOCX)

**S2 File. Response used for the determination of knowledge, practices and level of participation in mass drug administration at Erin-Ijesha and Ipole-Iloro communities in Southwest, Nigeria.**
(XLSX)

## Acknowledgments

The authors wish to acknowledge the support of the community leaders of Erin-Ijesha and Ipole-Iloro community for their support. We would also like to appreciate the state neglected tropical diseases (NTD) coordinators, local NTD coordinators and local mobilizers for their support towards ensuring the success of this study.

## Author contributions

**Conceptualization:** Oluwadamilare Ganiu Dauda, Akinlabi Mohammed Rufai, Monsuru Adebayo Adeleke.

**Formal analysis:** Oluwadamilare Ganiu Dauda, Oluwatoyin Adeola Oyeniran.

**Investigation:** Oluwadamilare Ganiu Dauda, Olusegun Quadri Adeshina, Abuh Ojomona Oboro, Azeezat Ayanbukola Ayannibi, Rhidollah Folashade Oyewusi, Glory Blessing Jokanola.

**Methodology:** Oluwadamilare Ganiu Dauda, Olabanji Ahmed Surakat.

**Supervision:** Akinlabi Mohammed Rufai, Kamilu Ayo Fasasi, Monsuru Adebayo Adeleke.

**Writing – original draft:** Oluwadamilare Ganiu Dauda, Akinlabi Mohammed Rufai, Olabanji Ahmed Surakat, Monsuru Adebayo Adeleke.

**Writing – review & editing:** Oluwadamilare Ganiu Dauda, Olabanji Ahmed Surakat, Kamilu Ayo Fasasi, Monsuru Adebayo Adeleke.

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
