## [Decision Letter · Decision Letter 0]

6 Mar 2025

Knowledge and attitude on blackflies and onchocerciasis and participation in Mass Drug Administration in frontline communities near Erin-Ijesha and Arinta waterfalls, Southwest Nigeria.

Dear Dr. Dauda,

Thank you for submitting your manuscript to PLOS Neglected Tropical Diseases. After careful consideration, we feel that it has merit but does not fully meet PLOS Neglected Tropical Diseases's publication criteria as it currently stands. Therefore, we invite you to submit a revised version of the manuscript that addresses the points raised during the review process. The editor agrees with one reviewer that the study as it is written now, is mostly of local interest. Therefore it is important to point out in a substantially revised version why the results are important for the national onchocerciasis elimination program in Nigeria and elsewhere in Africa.

Please submit your revised manuscript within 60 days May 05 2025 11:59PM. If you will need more time than this to complete your revisions, please reply to this message or contact the journal office at plosntds@plos.org. Please include the following items when submitting your revised manuscript:

We look forward to receiving your revised manuscript.

Kind regards,

Peter U Fischer

Academic Editor

Nigel Beebe

Section Editor

Shaden Kamhawi

co-Editor-in-Chief

Paul Brindley

co-Editor-in-Chief

**Journal Requirements:**

At this stage, the following Authors/Authors require contributions: Glory Blessing Jokanola. Please ensure that the full contributions of each author are acknowledged in the "Add/Edit/Remove Authors" section of our submission form.

3) Some material included in your submission may be copyrighted. According to PLOSu2019s copyright policy, authors who use figures or other material (e.g., graphics, clipart, maps) from another author or copyright holder must demonstrate or obtain permission to publish this material under the Creative Commons Attribution 4.0 International (CC BY 4.0) License used by PLOS journals. Please closely review the details of PLOSu2019s copyright requirements here: PLOS Licenses and Copyright. If you need to request permissions from a copyright holder, you may use PLOS's Copyright Content Permission form.

Potential Copyright Issues:

- Figure 1. Please provide a direct link to the base layer of the map (i.e., the country or region border shape) and ensure this is also included in the figure legend; and provide a link to the terms of use / license information for the base layer image or shapefile. We cannot publish proprietary or copyrighted maps (e.g. Google Maps, Mapquest) and the terms of use for your map base layer must be compatible with our CC BY 4.0 license.

4) In the online submission form, you indicated that "All data generated and/or analysed during this study will be made available by the corresponding author upon reasonable request". All PLOS journals now require all data underlying the findings described in their manuscript to be freely available to other researchers, either

- In a public repository

- Within the manuscript itself

- Uploaded as supplementary information.

**Reviewers' Comments:**

Reviewer's Responses to Questions

**Key Review Criteria Required for Acceptance?**

**Methods:**

-Are the objectives of the study clearly articulated with a clear testable hypothesis stated?

-Is the study design appropriate to address the stated objectives?

-Is the population clearly described and appropriate for the hypothesis being tested?

-Is the sample size sufficient to ensure adequate power to address the hypothesis being tested?

-Were correct statistical analysis used to support conclusions?

-Are there concerns about ethical or regulatory requirements being met?

Reviewer #1: (No Response)

Reviewer #2: -The broader methodology used in the study to achieve the objective are adequate. However, authors need to provide additional information around the population used for sample size calculation as well as information on the endemicity status of the LGA of the study communities.

Reviewer #3: -Are the objectives of the study clearly articulated with a clear testable hypothesis stated? Yes

-Is the study design appropriate to address the stated objectives? yes

-Is the population clearly described and appropriate for the hypothesis being tested? not clear how the study participants were selected. Not representative of the general population

-Is the sample size sufficient to ensure adequate power to address the hypothesis being tested? yes

-Were correct statistical analysis used to support conclusions? limited statistical analysis done

-Are there concerns about ethical or regulatory requirements being met? No

**Results:**

-Does the analysis presented match the analysis plan?

-Are the results clearly and completely presented?

-Are the figures (Tables, Images) of sufficient quality for clarity?

Reviewer #1: (No Response)

Reviewer #2: The analysis and data presentation in the current study are adequate.

Reviewer #3: -Does the analysis presented match the analysis plan? No analysis plan

-Are the results clearly and completely presented? Yes

-Are the figures (Tables, Images) of sufficient quality for clarity? Not very good

**Conclusions:**

-Are the conclusions supported by the data presented?

-Are the limitations of analysis clearly described?

-Do the authors discuss how these data can be helpful to advance our understanding of the topic under study?

-Is public health relevance addressed?

Reviewer #1: (No Response)

Reviewer #2: While the conclusions is largely supported by data presented, authors need to clarify, modify and delete some conclusion presented in the study as data presented in the study are inadequate to infer such conclusion. Conclusion should be within the limit of data presented and limitations of study should be included.

Reviewer #3: -Are the conclusions supported by the data presented? ok

-Are the limitations of analysis clearly described? Not described

-Do the authors discuss how these data can be helpful to advance our understanding of the topic under study? data are mainly of local importance

-Is public health relevance addressed? Yes

**Editorial and Data Presentation Modifications?**

Reviewer #1: (No Response)

Reviewer #2: (No Response)

Reviewer #3: "Comunities not previously mapped." What does this mean?

The study population was not representative of the general pululation. > 70% women, and young people underrepresented

Limited number of farmers generally most exposed to blackfly biting

No study limitations are mentioned

No recommendations are proposed to further investigate the onchocerciasis endemicity in the area?

Improving health education seems to be the only recommendation

No qualitative research done

No CDTI coverage data reported by CDDs?

**Summary and General Comments:**

Reviewer #1: (No Response)

Reviewer #2: This is generally a well done study. However, authors should address all the critical comments in the manuscript to ensure the interpretation of data and information generated from the study is not over-extended beyond what is necessary.

Reviewer #3: Results of the study are mainly of local importance

PLOS authors have the option to publish the peer review history of their article (what does this mean? ). If published, this will include your full peer review and any attached files.

**Do you want your identity to be public for this peer review?** For information about this choice, including consent withdrawal, please see our Privacy Policy .

Reviewer #1: No

Reviewer #2: No

Reviewer #3: **Yes: ** Robert Colebunders

**Figure resubmission:**

**Reproducibility:**



---

## [Editor Report · Decision Letter 1]

12 Jun 2025

Knowledge and attitude of populations on blackflies and onchocerciasis and participation in Mass Drug Administration in first-line communities near Erin-Ijesha and Arinta waterfalls, Southwest Nigeria.

Dear Dr. Dauda,

Thank you for submitting your manuscript to PLOS Neglected Tropical Diseases. After careful consideration, we feel that it has merit but does not fully meet PLOS Neglected Tropical Diseases's publication criteria as it currently stands. Therefore, we invite you to submit a revised version of the manuscript that addresses the points raised during the review process.

Please submit your revised manuscript within 60 days Jul 12 2025 11:59PM. If you will need more time than this to complete your revisions, please reply to this message or contact the journal office at plosntds@plos.org. Please include the following items when submitting your revised manuscript:

We look forward to receiving your revised manuscript.

Kind regards,

Peter U Fischer

Academic Editor

Nigel Beebe

Section Editor

Shaden Kamhawi

co-Editor-in-Chief

Paul Brindley

co-Editor-in-Chief

**Additional Editor Comments (if provided):**

The authors addressed most of the reviewers’ comments. However, the editor has still some concerns about this ms.

Line 82/83 Reference 10 clearly states that the reported population treated in Nigeria in 2023 was 34.9 million. What do the authors mean with ‘Currently, ivermectin has been administered to over 45 million Nigerians…’? [10]

Table 1. What do the authors mean with ‘Numbers’? Number of study participants?

Table 3: Onchodermatitis and itching are the most common signs of onchocerciasis. Why was this not included? How is ‘ocular dermatitis’ defined? Do the authors mean periorbital dermatitis? Ocular disease caused by O. volvulus that can be easily seen is sclerosing ceratitis. How would this be included in the table. How is blindness defined?

Line 199. Why did the authors introduce the phrase mass administration of medicine (MAM)? What is the difference to the commonly used term mass drug administration (MDA)?

Table 4. CDTi is a special form of MDA. What is the difference between CDTi and MAM?

Line 331. Why is knowledge of blackfly bioecology important to increase compliance to MDA? Many studies have shown that trust to community drug distributors is more important than disease specific knowledge. The authors may want to address this point more in detail in their discussion.

In addition, the ms still contains typos (for example Table 2 blackflies vs black flies).

**Journal Requirements:**

1) Some material included in your submission may be copyrighted. According to PLOSu2019s copyright policy, authors who use figures or other material (e.g., graphics, clipart, maps) from another author or copyright holder must demonstrate or obtain permission to publish this material under the Creative Commons Attribution 4.0 International (CC BY 4.0) License used by PLOS journals. Please closely review the details of PLOSu2019s copyright requirements here: PLOS Licenses and Copyright. If you need to request permissions from a copyright holder, you may use PLOS's Copyright Content Permission form.

Potential Copyright Issues:

i)Figure 1. Thank you for including this link: https://www.hydrosheds.org/products/hydrorivers . However, it does not comply with PLOS CC:BY 4.0 requirements.

Please (a) provide a direct link to the base layer of the map (i.e., the country or region border shape) and ensure this is also included in the figure legend; and (b) provide a link to the terms of use / license information for the base layer image or shapefile. We cannot publish proprietary or copyrighted maps (e.g. Google Maps, Mapquest) and the terms of use for your map base layer must be compatible with our CC BY 4.0 license.

2) We note that your Data Availability Statement is currently as follows: "All data generated and/or analysed during this study is uploaded as supplementary information.". Please confirm at this time whether or not your submission contains all raw data required to replicate the results of your study. Authors must share the “minimal data set” for their submission. PLOS defines the minimal data set to consist of the data required to replicate all study findings reported in the article, as well as related metadata and methods (https://journals.plos.org/plosone/s/data-availability#loc-minimal-data-set-definition).

**Reviewers' Comments:**

**Figure resubmission:**
---

## [Editor Report · Decision Letter 2]

31 Jul 2025

Dear Mr Dauda,

We are pleased to inform you that your manuscript 'Knowledge and attitude of populations on blackflies and onchocerciasis and participation in Mass Drug Administration in first-line communities near Erin-Ijesha and Arinta waterfalls, Southwest Nigeria.' has been provisionally accepted for publication in PLOS Neglected Tropical Diseases.

Best regards,

Nigel Beebe, PhD

Section Editor

Nigel Beebe

Section Editor

Shaden Kamhawi

co-Editor-in-Chief

Paul Brindley

co-Editor-in-Chief

Thanks for attending to the reviewer's comments.

---

## [Editor Report · Acceptance letter]

Dear Mr Dauda,

We are delighted to inform you that your manuscript, "Knowledge and attitude of populations on blackflies and onchocerciasis and participation in Mass Drug Administration in first-line communities near Erin-Ijesha and Arinta waterfalls, Southwest Nigeria.," has been formally accepted for publication in PLOS Neglected Tropical Diseases.

Best regards,

Shaden Kamhawi

co-Editor-in-Chief

Paul Brindley

co-Editor-in-Chief
